# Aerodynamic Characteristics and Lateral Displacements of a Set of Two Buildings in a Linked Tall Building System

**DOI:** 10.3390/s21124046

**Published:** 2021-06-11

**Authors:** Zengshun Chen, Bubryur Kim, Dong-Eun Lee

**Affiliations:** 1School of Civil Engineering, Chongqing University, Chongqing 400045, China; zengshunchen@cqu.edu.cn; 2Department of Architectural Engineering, Dong-A University, Busan 49315, Korea; 3Department of ICT Integrated Ocean Smart Cities Engineering, Dong-A University, Busan 49315, Korea; 4School of Architecture, Civil, Environment and Energy Engineering, Kyungpook National University, 80, Daehak-ro, Buk-gu, Daegu 41566, Korea; dolee@knu.ac.kr

**Keywords:** linked tall building, particle image velocimetry, pressure measurement, staggered arrangement, wind-induced response

## Abstract

This study evaluates the aerodynamic characteristics and lateral displacements of two staggered buildings in a linked-building (LB) system. Particle image velocimetry and pressure measurements are employed, and the lateral displacement is evaluated using a 3-dimensional analytical model. When the gap distance between two non-linked buildings is small, the wind flows in a narrow jet, and a strong suction is generated on the inner surfaces of the two buildings, leading to a large cross-wind-induced response. However, the cross-wind-induced response is significantly reduced when a link is installed, because the suction forces generated from the buildings are in opposite directions and have a negative aerodynamic correlation. Conversely, with a large gap distance, the buildings at the front obstruct the wind blowing toward the rear buildings. Therefore, while the pressure distribution, wind-force coefficients, and wind-induced responses of the front and rear buildings show similar trends, the magnitude of impact on the front building is larger than that on the rear building. Installing a link is demonstrated to reduce the wind-induced response of the buildings in an LB system. However, the reduction in the along-wind-induced response is less than that in the cross-wind-induced response when the gap distance is small.

## 1. Introduction

High-rise buildings can lead to windy conditions over large surrounding areas owing to their huge sizes; furthermore, they are more affected by wind loads than are low-rise buildings because of their slenderer structures. Hence, response and wind-resistant design are necessary when designing tall buildings. In a linked-building (LB) system, such as skybridges, skypools, and skygardens, where a link connects two tall buildings, the two buildings are very close to each other. Therefore, the wind-load components acting on them are more complex. Moreover, the wind-induced lateral displacements of the two tall buildings affect each other. Consequently, it is more complex to design an LB system to resist wind loads than it is for a typical standalone high-rise building.

Several tall buildings are densely concentrated in modern cities, and they are near each other. The wind-induced response and aerodynamics, such as wind velocities and vectors around the buildings, wind pressures on surfaces, and wind loads, vary significantly among tall standalone structures [1,2,3,4,5,6,7]. Therefore, researchers have studied the wind velocities and vectors around numerous buildings using flow visualization or particle image velocimetry (PIV) [8,9,10]. Moreover, several previous studies have examined the characteristics of wind-load components generated in areas with dense concentrations of buildings [4,5,6,7,8,9,10,11,12,13,14,15,16,17,18,19,20,21]. However, while some have studied the consequences of adjacent tall buildings on the wind-force components of the principal tall building, others have investigated the wind-force components related to the principal tall building and not of the adjacent tall buildings. In an LB system, the characteristics of the wind forces at both buildings must be investigated. Furthermore, it is necessary to investigate how the wind-force components generated at each building impact and interact with the other buildings.

Previous studies have considered the characteristics of the wind-load components presented for two buildings with an LB system. Researchers have discussed the aerodynamic characteristics of wind-load components caused by the gap distance between two tall buildings in an LB system [22] and their variations depending on the location of the link [23]. The characteristics of wind loads generated by the two buildings in an LB system were studied in detail using orthogonal decomposition [24]. PIV was also used to examine the wind velocities and vectors generated around the two buildings with an LB system [20]. When an LB system is constructed in the form of a tall building with square sections, the wind-load components acting on the tall buildings are large, owing to vortex shedding. Therefore, researchers have studied the use of circular sections as a method to reduce these wind loads [21]. However, the LB systems in these previous studies were mainly arranged side by side, and some of the studies were performed on buildings in a tandem arrangement. Research on staggered arrangements is sparse. Moreover, several studies on staggered arrangements investigated and emphasized the aerodynamic characteristics of cylinders rather than typical buildings [25,26,27,28,29,30,31]. As mentioned study of the wind-load characteristics of actual LB systems has been neglected. Further investigation is needed concerning wind loads alongside the aerodynamic characteristics of LB systems.

Research has been conducted on the lateral displacement of LB systems. For instance, the modeling of structural coupling used to identify the impacts of the interactions between two buildings has been examined [32]. Moreover, the dynamic characteristics of an LB system have been assessed using a building model [33]. Additionally, the aerodynamic correlation and lateral displacements of an LB system have been evaluated using proper orthogonal decomposition [34]. However, these studies did not address staggered building arrangements.

This study examines the aerodynamic characteristics (e.g., wind velocities and vectors around the system, pressure distribution, and wind-force coefficients) of two staggered buildings in an LB system and evaluates the wind-induced response (lateral along-wind-induced and lateral cross-wind-induced) and the impact of the link on the lateral displacements. Wind flow and pressure were measured with a PIV test and a multi-pressure measurement system (MPMS), respectively. The wind loads were obtained from the wind-pressure data. Subsequently, the obtained wind loads and a 3D analytical model were used to evaluate the lateral displacements of two tall buildings with and without a link.

## 2. Materials and Methods

A wind-tunnel experiment was conducted to understand the aerodynamic characteristics generated by the LB system and obtain the wind loads. The height of the two buildings was assumed to be 160 m, and both buildings have the same square cross section. The width of the cross section was assumed to be 30 m to reflect the characteristics of a typical LB system. This study also assumed that the link was installed at the top floor of the two tall buildings of the LB system. Two wind-tunnel experiments were conducted separately using the PIV test and MPMS.

The gap between the two tall buildings plays a crucial role in various wind velocities, vectors, and wind-force components. Hence, this study considered wind-tunnel experiments on models with different gap ratios (S/B), where S denotes the spacing between the two tall buildings, and B denotes the building breadth, as shown in Figure 1. This study examines the wind velocities and vectors, pressure distributions, wind loads, and wind-induced responses when the gap ratio is relatively large (e.g., S/B = 1.5) or small (e.g., S/B = 0.33). To replicate the two tall buildings in a staggered arrangement, the wind direction was from the left side of the building front, and the angle α was set to 40°, as in Figure 1.

Furthermore, this study defined the surface receiving wind from the front as the windward surface, and the surface blocked from the wind was designated as the leeward surface. The surfaces of the two buildings facing each other and those facing outward were designated as the inner and outer surfaces, respectively. The buildings on the left and right were designated as Buildings 1 and 2, respectively, as shown in Figure 1.

### 2.1. PIV

Wind velocities and vectors are crucial for evaluating the wind environment around two tall buildings as well as the wind-force components and wind-induced responses. Hence, the horizontal wind velocities and vectors around the LB models were evaluated using PIV. Even in a staggered LB system, PIV can be used to measure factors, such as the wind velocities and vectors around the LB models, to improve the comprehension of the aerodynamics and causes of wind-force components and lateral displacement.

For the wind-tunnel experiment, the LB models were downscaled 300 times during manufacturing. The wind-tunnel laboratory of the Shimizu Corporation in Japan was used for the PIV test. The wind tunnel used for laboratory testing was 3.5-m wide and 2.5-m deep, which was sufficient to measure the wind velocities and vectors around the downscaled LB system. The mean wind speed at the top of the building model was measured at 7.2 m/s, the Reynolds number was approximately $5.2\times 10^{4}$, and the sampling frequency for measuring was 150 Hz. Figure 2 shows the LB model and equipment used for measuring the wind flow.

The camera for measuring the horizontal wind velocities and vectors was installed on the LB model. The measurements at three different elevations, as shown in Figure 3, were performed considering the height of the building models (H) and measurement elevation (h). At 0.8 H, h/H = 0.8, the wind speed was high, and the interference effects occurring in both buildings were most apparent [22]. As such, 0.8 H is crucial for identifying the overall wind velocities and vectors around the two tall buildings with the LB system. Moreover, 0.3 H, h/H = 0.3 values correspond to the lowest elevation of the LB model that can be evaluated using the PIV measurement system. Thus, the wind flow at the lower part of tall buildings can be investigated. The speed at which the wind blows in the wind tunnel was weaker at other elevations.

At 0.8 H and 0.3 H, there was no link connecting the two tall buildings. However, the link connecting the two tall buildings at the top covers the entire gap. Consequently, a camera for evaluating the wind velocities and vectors from the top of the tall building model cannot be used to measure the wind flow in the gap. Therefore, the link was removed to measure the wind velocities and vectors in the gap at 0.8 H and 0.3 H. Indeed, the link impacted the aerodynamic characteristics of the LB system only locally around the link installation; at 0.8 H and 0.3 H where the distance was relatively far from the installation, it had little impact on the aerodynamic characteristics, and the wind flow could still be reasonably evaluated [23].

The wind velocities and vectors related to the two tall buildings were evaluated at 0.8 H and 0.3 H. Conversely, at 1.0 H (h/H = 1.0), the wind-flow patterns were expected to differ from those at other elevations because of the link, as shown in Figure 3. Therefore, this fact was studied further. At 1.0 H, most of the wind velocities and vectors around the two buildings and the link could be measured.

### 2.2. MPMS

The results obtained from the PIV test provide information for identifying the wind flow around the two tall buildings with an LB system. However, to evaluate wind-induced responses, wind loads should be determined. The wind pressure that may occur on the building surface can be identified through MPMS, which is then used to calculate wind loads, such as along-wind, cross-wind, and tensional forces.

As mentioned, the link in an LB system only locally affects the aerodynamics of the location where the link is installed and not the entire LB system [23]. Hence, in MPMS, the wind loads were obtained considering only the two buildings with the link removed, and wind-induced responses were calculated. Building models with a scale of 1:400 were created for the wind-tunnel experiments. Accordingly, each model was built with a size of 7.5 × 7.5 × 40 cm. MPMS measurements were organized in the boundary layer wind tunnel of the Wind/Wave Tunnel Facility at the Hong Kong University of Science and Technology. Figure 4 shows that the wind-pressure sensors were installed on the tall building models.

Figure 5 shows that the MPMS sensor in this wind-tunnel experiment consisted of nine layers with 20 wind-pressure measurement sensors installed at each layer. A number of 360 wind-pressure sensors were installed in the LB system with 45 wind-pressure sensors on each surface. The wind loads are higher in the upper part of the buildings where the interference effects are most apparent [22]; hence, the wind-pressure sensors were installed in the upper part with smaller gaps than at the lower part. In previous studies, the wind-pressure data from each wind-pressure sensor were used to examine the along-wind, cross-wind, and torsional-wind forces of the two buildings [35,36]. This study also did so. At the top of the two-building LB models, the mean wind speed was 16 m/s. The wind pressure of the buildings was measured at the receiving frequency of 500 Hz for each wind-pressure sensor. Moreover, the Reynolds number was $8\times 10^{4}$.

The numerical compensation approach was applied to correct the tubing approaches [37]. The data acquired from the MPMS were expressed using fast Fourier transformation, and the Fourier series was designed through the frequency response and phase shift of the MPMS. Additionally, the proper results were obtained using the inverse Fourier transformation.

Figure 2 shows the setup of the PIV test that was conducted using a staggered arrangement. The laser was fired at an angle to observe the wind flow in as many parts as possible. For pressure measurement, a photo was taken (Figure 4) to ensure that the pressure taps on the surface and conditions of the wind-tunnel test could be properly confirmed. All tests were performed in a staggered arrangement as shown in Figure 1.

Figure 6 shows that the mean wind speed follows a power law function with an exponent of 0.2, indicating an open terrain determined by the AS/NZS 1170.2:2002 standards. In the figure, H is the height of the tall building models of the LB system, and z is the measurement height of the wind profile. The PIV test also shows similar wind profiles following a power law function with an exponent of 0.2, indicating an open terrain. This study did not consider any particular place or time, and an open terrain was considered. In terms of wind load, in MPMS, the wind speed blowing at the top of the model during testing was 16 m/s, which corresponds to 43 m/s for an actual building. The influence of relatively large wind load was evaluated. This study covers wind load and does not evaluate seismic load. As such, additional research is needed on seismic load. The mean wind speed (i.e., 42.8 m/s) used to calculate structural responses has a return period of 50 years. Hence, this wind speed is sufficient for evaluating structural safety. The wind-load results obtained from the wind-tunnel test may differ from actual situations. Nevertheless, this test was performed in accordance with the AS/NZS 1170.2:2002 standards and is applicable to general real-life situations.

In the PIV test, strong wind speed in the wind tunnel caused the cameras to vibrate, preventing accurate measurement of the wind flow. Therefore, a relatively weak wind speed (7.2 m/s) was considered. In contrast, as the MPMS does not require a camera, the building model can be constructed such that the vibrations caused by strong winds can be prevented. Therefore, a relatively strong wind speed (16 m/s) was employed. The two experiments were conducted separately under different conditions. However, the results obtained from the PIV test can provide referential information to compare with those from the MPMS.

## 3. Aerodynamic Characteristics

### 3.1. Wind Flow around the LB System

This study conducted a flow visualization to examine the wind characteristics around an LB system comprised of two staggered buildings. Accordingly, through PIV measurements, the instantaneous wind flows (velocity and vector) were measured around the LB system for two cases with different gap distances between the two tall buildings. Subsequently, the mean velocity and mean vectors were obtained. This section examines the characteristics of wind flow around the two buildings of the LB systems using the mean wind velocities and mean vectors. Wind flow can help us understand the pressure patterns on the surfaces of the LB system and the wind loads (e.g., along-wind and cross-wind forces) to facilitate the understanding of wind-induced responses.

Figure 7 and Figure 8 show the horizontal wind flows, including mean velocity magnitudes and mean velocity vectors around the LB system, for three different elevations and two different gap distances. Figure 1 shows the wind direction from the left front side of the buildings in the LB system. The white rectangles in Figure 7 and Figure 8 represent the cross sections of the LB model. To understand the wind velocity magnitudes better, the *x*- and *y*-axes were spaced according to the size of one of the buildings’ cross section. As shown in Figure 6, the wind speed at 0.8 H was higher than that at 0.3 H; accordingly, in Figure 7a,b, the mean wind velocity around the building was greater at 0.8 H than at 0.3 H. Moreover, phenomena such as channeling and shear layers were more apparent at 0.8 H. These results predict that the wind loads or dynamic characteristics will be clear at 0.8 H in an LB system with a staggered arrangement.

With small gap distance, as shown in 0.3 H and 0.8 H in Figure 7a,b, the mean velocity between the two buildings is slightly larger, owing to the channeling phenomenon between the inner surfaces of the two buildings; thus, a narrow jet of fluid flows. Moreover, the two tall buildings behave as a single bluff-body, because vortex shedding occurs only at the outer shear layers. Conversely, when the gap distance is large (e.g., 0.3–0.8 H) the magnitudes of the mean velocities at the outer and inner surfaces of the two tall buildings are large, owing to shear layers from the windward side of the two tall buildings. Despite the large gap distance, Building 2 was partially blocked from the blowing wind by Building 1, and the shear layers generated by Building 1 partially affect Building 2.

Furthermore, a large gap ratio between the two tall buildings at S/B = 1.5 causes the mean velocity between the two tall buildings to become larger than when the gap ratio is small. Additionally, the area with high mean velocity increases at the rear of the two tall buildings, and this area increases further when the gap ratio is large because when the gap ratio is large, the shear layers generated inside Building 1 can increase considerably without obstruction, and these increased shear layers also affect Building 2 in the LB system.

At 1.0 H with a link at the top (Figure 7c), the mean velocity is large owing to shear layers occurring only on the outer surfaces of the two buildings. Furthermore, the mean velocity of the part blocked from the blowing wind by the two buildings and the link connecting them is smaller than that at the other parts.

The wind vector around the two tall buildings of the LB system was also examined using the mean vectors shown in Figure 8. With a small gap distance, an inherently biased vector in the gap area exists between the two tall buildings. The biased vector patterns shift from one building to the other and back continually [38,39,40]. Because the models in this study were two tall, staggered buildings, the pattern generally shifted to one side. As the gap distance increases, the wind vector patterns become more complex, and vortex formation and shedding can occur at the inner shear layers of Building 1. Consequently, owing to the wind flow at Building 1, the vortex shedding in the gap area between the two buildings affects a wide area, including Building 2. Moreover, the inner shear layers of Building 1 merge with those of Building 2. The combined inner shear layers are predicted to contribute to the formation of complex separation bubbles and turbulent reattachment.

Figure 8 shows that the wind-flow patterns occur because of large-scale vortices, such as recirculation zones in the wake region at the rear of the building models, formed by shear layers of the LB system. These results provide visual information through the PIV data on the degree to which the wake region and shear layers behind the buildings affect the surroundings of the tall buildings. With a small gap distance, a wake fluctuation is observed behind the leeward surfaces of both buildings and the outer surface of Building 2. One is also observed behind the leeward and inner surfaces of Building 1 when the gap distance is large. However, the observations are only partial when the gap distance is large, owing to the limited range of camera measurements on the leeward and outer surfaces of Building 2. Furthermore, wake fluctuation is expected to induce suction on each surface while contributing to small negative pressure distributions.

### 3.2. Pressure Distribution and Wind Loads on the LB System

It is meaningful to understand the characteristics of wind-force components before evaluating the lateral displacement of the LB system or the effect of the link in the LB system on the wind-induced response. These wind loads are related to pressure distribution and wind-force coefficients. For applying the data obtained from the pressure measurement to the prototype, a non-dimensional wind-pressure coefficient at the pressure sensor (i, j) on each surface of the LB model is necessary and can be determined as
(1)$$C_{\mathrm{ij}}^{S}\left(t\right)=\frac{p_{\mathrm{ij}}^{s}\left(t\right)-p_{0}}{\frac{1}{2}{\rho V}_{H}^{2}}$$
where i is the pressure sensor level, j is the pressure sensor number on each level, s is the notation of the particular surface (windward, leeward, outer, and inner surfaces), $p_{\mathrm{ij}}^{s}$ is the measured pressure at the pressure sensor (i and j) on each surface s, p_0_ is the local static pressure, ρ is the air density during wind-tunnel testing, and V_H_ is the wind velocity magnitude at the top of the LB system.

Figure 9 and Figure 10 show the mean pressure distribution of the two tall buildings with an LB system, considering two different gap distances. When the wind direction is 40°, a positive pressure distribution is observed on the windward and outer surfaces of Building 1 regardless of the gap distance between the two tall buildings. In contrast, in Building 2, the positive pressure distribution of the windward and inner surfaces is observed with a large gap distance. The wind flows in Figure 7 and Figure 8 indicate that even when the gap distance is large, Building 1 blocks Building 2 from the wind. Therefore, the positive pressures of the windward and inner surfaces have smaller magnitudes than those of Building 1.

With a small gap distance, the negative pressure value on the inner surface of Building 2 in the area near the windward edge of the two buildings is very large. Thus, with a small gap distance, channeling between the two buildings causes a fully developed flow to approach the gap, resulting in negative pressure, as shown in Figure 7 and Figure 8. This channeling effect also contributes to the negative pressure distribution on the inner surfaces of both buildings with a small gap distance.

Conversely, when the gap ratio is large, a relatively weak negative pressure distribution is observed on the inner surface of Building 1. The negative pressure occurring on the inner surface of Building 1 is related to wake fluctuation rather than the effect of channeling and the narrow jet of fluid flowing between the two buildings, as shown in Figure 7 and Figure 8. The mean pressure distributions on the other surfaces are less affected by the gap ratio because these surfaces only experience wake fluctuations caused by the wind.

Because wind-force components lead to generalized forces and wind-induced lateral displacements, the distribution of the wind-force components, which are the mean values of the wind-force components, should be mentioned. The wind-force component coefficients, $C_{D,i}$ (along-wind-force component coefficient) and $C_{L,i}$ (cross-wind-force component coefficient), are defined as
(2)$$C_{D,i}=\frac{F_{D,i}}{\frac{1}{2}{\rho V}_{H}^{2}B\Delta h}{\;\;\;C}_{L,i}=\frac{F_{L,i}}{\frac{1}{2}{\rho V}_{H}^{2}B\Delta h}\;$$
where $F_{D,i}$ and $F_{L,i}$ are the along-wind and cross-wind-force components, respectively, at height i and Δh is related to tributary height. Figure 11 shows the mean values of the wind-force component coefficients acting on the two tall buildings of the LB system when the gap distances are large and small. 

The along-wind-force coefficients are not impacted by the gap distance in Building 1 or 2, and the results obtained with different gap distances are similar, as shown in Figure 11. However, the along-wind-force coefficients of Building 2 are smaller than those of Building 1. Thus, as shown in the wind flow in Figure 7 and the pressure distribution in Figure 10, Building 1 blocks the wind blowing to Building 2 partially, thereby reducing the along-wind-force coefficients of Building 2.

Moreover, the cross-wind-force coefficients are significantly impacted by the gap distance. In the case of Building 1, with a small gap distance, suction is observed on the inner surface, as shown in Figure 9, increasing the cross-wind-force coefficients. Similarly, in the case of Building 2, with a small gap distance, the cross-wind-force coefficients are reduced owing to suction generated on the inner surface. Unlike the cross-wind-force coefficients under a large gap distance, it has negative values. Conversely, when the gap distance is large, because Building 2 is blocked from the wind, the cross-wind-force coefficients are smaller than those of Building 1; similar trends are observed with the cross-wind-force coefficients of Building 1.

The experimental method applied in this study is an accurate technique used for evaluating wind loads generated from buildings. The test results were compared to the results of similar prior research, and as described above, the phenomenon of wind load is logically explained. Therefore, the results of this test are reliable. The wind-tunnel test results should be regarded as more reliable than the simulation, and any result that cannot be confirmed through the wind-tunnel test must be examined via simulation. Nevertheless, as it is time-consuming to perform a complete simulation, such results will be covered in a separate paper.

## 4. Lateral Displacements of LB System

### 4.1. Analytical Model of LB System

The prototype dimensions and structural properties of the two tall buildings in the LB system were assumed to be identical. Figure 12 shows that each building consists of a frame-core wall structure. This study also assumed that the link was installed at the top floor of the two tall buildings. A 3D analytical model was developed, assuming that each building in the LB system had 40 floors. A height of 4 m per floor was assumed.

Because lateral displacements caused by wind-force components are dominant in tall buildings, vertical deformations were omitted. The structural properties of the link were normalized according to the structural properties of the two tall buildings as follows:(3)$$\lambda_{m}=\frac{m_{\mathrm{link}}}{m_{\mathrm{building}}}{\;\;\;\lambda }_{\mathrm{ka}}=\frac{k_{a}}{k_{e,b}}{\;\;\;\lambda }_{\mathrm{kb}}=\frac{k_{b}}{k_{e,\;t}}{\;\;\;\lambda }_{h}=\frac{h_{\mathrm{link}}}{h_{\mathrm{builidng}}}$$
where m_link_ is the link mass, and m_building_ is the entire building mass. Moreover, h_link_ is the link installation location, and h_building_ represents the height of the two tall buildings in the LB system. k_a_ is the axial stiffness of the link of the LB system. k_b_ is the bending stiffness of the link of the LB system. k_e,b_ is the lateral stiffness of the two buildings. k_e,t_ is torsional stiffness of the two buildings. The normalized mass $\lambda_{m}$, axial stiffness $\lambda_{\mathrm{ka}}$, bending stiffness $\lambda_{\mathrm{kb}}$, and link elevation $\lambda_{h}$ were 0.01, 10, 5, and 1.0, respectively. The link was designed assuming that it connects the two buildings tightly.

Wind loads appear in various forms, including along-wind, cross-wind, and torsional-wind loads. The 3D analytical model must be developed considering these loads, and the wind-induced response obtained from the analytical model can appear in various directions. The equation of motion of the 3D analytical model of the LB system is as follows:(4)$$\left[M_{B}+M_{L}\right]\ddot{D}+C_{B}\dot{D}+\left[K_{B}+K_{L}\right]D=F$$
where **M_B_**, **C_B_**, and **K_B_** are the mass, damping, and stiffness matrices of each tall building in the LB system, respectively. **M_L_** is the additional link mass. **K_L_** is the link stiffness matrices. F = $\left\{F_{1x},F_{1y},F_{1\theta },F_{2x},F_{2y},F_{2\theta }\right\}$ is the time history of the wind-force components from the MPMS, where F_gs_ is the wind-force components on building g (Building 1 or Building 2) in the direction s (wind direction x, y, and θ, where x, y, and θ are the cross-wind, along-wind, and torsional-wind forces, respectively). $\ddot{D},\;\dot{D}$, and D are the lateral acceleration, velocity, and displacement responses, respectively. **M_B_**, **C_B_**, and **K_B_** can be expressed as follows:(5)$$M_{B}={\left[\begin{array}{cc} M_{1} & 0\\ 0 & M_{2} \end{array}\right]}_{6m\times 6m}$$
(6)$$C_{B}={\left[\begin{array}{cc} C_{1} & 0\\ 0 & C_{2} \end{array}\right]}_{6m\times 6m}$$
(7)$$K_{B}={\left[\begin{array}{cc} K_{1} & 0\\ 0 & K_{2} \end{array}\right]}_{6m\times 6m}$$
(8)$$M_{1}=M_{2}={\left[\begin{array}{ccc} M_{x} & 0 & 0\\ 0 & M_{y} & 0\\ 0 & 0 & J \end{array}\right]}_{3m\times 3m}$$
(9)$$K_{1}=K_{2}={\left[\begin{array}{ccc} K_{\mathrm{xx}} & 0 & K_{x\theta }\\ 0 & K_{\mathrm{yy}} & K_{y\theta }\\ K_{\theta x} & K_{\theta y} & K_{\theta \theta } \end{array}\right]}_{3m\times 3m}$$
where m represents the number of floors, $M_{1}$ and $M_{2}$ are the mass matrices, $C_{1}$ and $C_{2}$ are the damping matrices, and $K_{1}$ and $K_{2}$ are the stiffness matrices of Buildings 1 and 2, respectively. $M_{x}$ and $M_{y}$ are the structural mass of the two tall buildings. J is the mass moment of the inertia sub-matrices of the two tall buildings. The stiffness matrix $K_{L}$ of the link is set for the LB system as follows:(10)$$K_{L}=\left[\begin{array}{cccccc} k_{a} & 0 & 0 & -k_{a} & 0 & 0\\ 0 & \frac{12k_{b}}{l^{2}} & \frac{6k_{b}}{l} & 0 & \frac{-12k_{b}}{l^{2}} & \frac{6k_{b}}{l}\\ 0 & \frac{6k_{b}}{l} & \left(4+\beta \right)k_{b} & 0 & \frac{-6k_{b}}{l} & \left(2-\beta \right)k_{b}\\ -k_{a} & 0 & 0 & k_{a} & 0 & 0\\ 0 & \frac{-12k_{b}}{l^{2}} & \frac{-6k_{b}}{l} & 0 & \frac{12k_{b}}{l^{2}} & \frac{-6k_{b}}{l}\\ 0 & \frac{6k_{b}}{l} & \left(2-\beta \right)k_{b} & 0 & \frac{-6k_{b}}{l} & \left(4+\beta \right)k_{b} \end{array}\right]$$

The symbols mentioned above are related to the link properties of the LB system; l is the link length. The axial stiffness k_a_ and bending stiffness k_b_ are expressed as EA/l and E$I_{b}$/l, respectively. $I_{b}$ is the moment of inertia, expressed as I/(1 + β). β is the shear deformation constant. A is the cross-section area. I is the moment of inertia. E is the Young’s modulus. 

The structural modal parameters of the LB system can be calculated using the following equation: (11)$$\left(K_{total}-\omega^{2}M_{total}\right)\Phi =0$$
where $K_{total}=K_{B}+K_{L}$ and $M_{total}=M_{B}+M_{L}$ are the stiffness and mass matrices of the entire LB system, respectively, ω is the natural frequency of the entire system, and **Φ** = {Φ_1x_, Φ_1y_, Φ_1z_, Φ_2x_, Φ_2y_, Φ_2z_} is the corresponding structural mode shape. Φ_gr_ is the structural mode shape of tall Building g (Building 1 or 2) in the r direction (x, y_,_ or z).

By using the mode superposition method, Equation (4) can be expressed as follows: (12)$${\ddot{q}}_{j}\left(t\right)+2\xi_{j}\omega_{j}{\dot{q}}_{j}\left(t\right)+\omega_{j}^{2}q_{j}\left(t\right)=\frac{F_{j}^{*}\left(t\right)}{M_{j}^{*}}$$
where q_j_ is the coordinate, ω_j_ is the natural frequency, and ξ_j_ is the damping ratio. F_j_^*^ is the wind force, and M_j_^*^ is the mass. These values are related to the j-th mode. F_j_^*^ and M_j_^*^ can be expressed in detail through the following equations: (13)$${\;F}_{j}^{*}=\overset{m}{\underset{i=1}{\sum }}\underset{s=x,y,\theta }{\sum }\underset{g=1,2}{\sum }F_{\mathrm{sg},\;t}\left(z_{i}\right)\varphi_{\mathrm{sg},j}\left(z_{i}\right),$$
(14)$$m_{j}^{*}=\sum_{i=1}^{m}\sum_{g=1,2}\left\{m\left(z_{i}\right)\left[\varphi_{xg,j}^{2}\left(z_{i}\right)+\varphi_{yg,j}^{2}\left(z_{i}\right)\right]+J\left(z_{i}\right)\varphi_{\theta g,j}^{2}\left(z_{i}\right)\right\}$$
where F_sg_(z_i_) is the wind-force time history obtained from the wind-tunnel testing, and z_i_ is the height of the i-th floor of the LB system. Φ_gr,j_ is the element of the j-th structural mode shape of Building g (1 or 2) in the direction r (x, y, or z).

As shown in Equation (12), the integration method was used for q_j_(t), the normal coordinate. The time-history wind-induced response can be calculated using the normal coordinate and Equation (15).
(15)$$D_{\mathrm{gs}}\left(z,t\right)=\underset{j}{\sum }D_{\mathrm{gs},j}=\underset{j}{\sum }q_{j}\left(t\right)\times \varphi_{\mathrm{gs},j}\left(z\right)$$
where $D_{\mathrm{gr}}$ is the j-th modal displacement of Building g (1 or 2) in the direction r (x, y, or z).

In summary, the wind-pressure coefficient can be obtained using Equation (1) and converted to the wind-force component coefficients, $C_{D,i}$ (along-wind-force component coefficient) and $C_{L,i}$ (cross-wind-force component coefficient), using Equation (2). This converted wind-force time history can be calculated as the time-history wind-induced response through Equations (13)–(15). The accuracy of the 3-dimensional (3D) analysis model mentioned in this study has been validated in [33]. However, the previous study presented the results of 3D analysis and wind-tunnel tests using a side-by-side arrangement of two buildings. However, this study further investigated the staggered arrangement.

For the first natural frequency of a single building in the LB system, a mode shape formed in the side direction of the building with a value of 0.244 Hz. For the second natural frequency, a mode shape formed in the front direction of the building with the same value as the first natural frequency (0.244 Hz). For the third natural frequency, a mode shape formed in the rotational direction of the building with a value of 0.411 Hz. In an LB system where a link connects two buildings, the dynamic characteristics of the system varied with the location of the link and the structural properties.

### 4.2. Standard Deviations of Lateral Displacement

Figure 13, Figure 14, Figure 15 and Figure 16 show the standard deviations of the lateral displacements according to the time history in the two buildings caused by wind loads at different gap distances with and without the link. The standard deviation of lateral displacement was employed to reflect the maximum lateral displacement when assessing the safety of the buildings against wind loads. However, because the peak displacements were generated temporarily, they did not reflect the overall wind-induced response according to the time history, and the standard deviation was used instead to reflect the wind-induced response over time.

As displayed in Figure 11, the along-wind-force coefficients are larger for Building 1; therefore, the effects of the lateral displacement of Building 1 are larger in the absence of a link, as shown in Figure 13 and Figure 14. However, when a link is installed, the along-wind-force coefficients are reduced in most cases except with a small gap distance.

Apart from the no-link case, when a link is installed, the lateral displacements in the along-wind and cross-wind directions are different. However, when there is a link, the along-wind responses of both buildings are similar because the link between the two tall buildings in an LB system connects the two buildings; thus, the wind responses of both buildings are similar.

Figure 15 and Figure 16 show the cross-lateral displacement that present similar results to the along-wind-induced responses. However, compared with the along-wind responses, the cross-wind response of Building 1 was reduced to a greater extent than that of Building 2 when a link was present. Moreover, in the cases of large and small gap distances, the wind-force coefficients generated in Building 1 are higher than those in Building 2. Hence, Building 2 serves the role of offsetting Building 1, revealing a relatively large wind-induced response. Therefore, the cross-wind response of Building 1 in the LB system is highly reduced compared with the case where no link was included.

In particular, as indicated by the standard deviation of the crosswind-induced response of Building 1 in Figure 15a, the wind-induced response in the absence of a link is approximately 93 mm with a small gap distance. With a large gap distance, as shown in Figure 16a, the standard deviation of the lateral displacement is approximately 87 mm. With a small gap distance, the lateral displacement is large in the absence of a link. However, in the presence of a link, the wind-induced response is small when there is a small gap distance.

These results reveal that the link generally plays a crucial role in reducing the lateral displacement of the LB system. However, the effect varies with the gap distance, as the gap distance affects the aerodynamic characteristics acting on the LB system and the aerodynamic correlation between the two tall buildings. Owing to the link effect on the LB system, the lateral displacement reduction is closely related to the aerodynamic correlation between the buildings. Additional studies are necessary to assess this aerodynamic correlation, as discussed in the following section.

### 4.3. Trajectories for Lateral Displacement

To assess the aerodynamic correlation between the two buildings, this study analyzed the along-wind and crosswind-induced lateral displacement according to the time history through the trajectories, as shown in Figure 17 and Figure 18.

With a small gap distance and no link, the cross-wind lateral displacement of Building 1 is large and positive in one direction, as shown in Figure 17. Conversely, the cross-wind lateral displacement of Building 2 is negative and biased toward one direction, as indicated by the mean values of the local wind-force component coefficients in Figure 11. With a small gap distance, the cross-wind-force coefficients are large because of the suction occurring on the inner surface; the cross-wind response is also affected. Furthermore, the suction forces at the inner surfaces of both buildings occur in opposite directions. Therefore, the cross-wind-force coefficients and wind-induced lateral displacements also occur in opposite directions. Moreover, the cross-wind response of Building 1 is more biased toward one side compared with that of Building 2. The suction at the inner surface of Building 1 is more apparent and larger, as shown in Figure 10. Consequently, the cross-wind-force coefficients and wind-induced lateral displacements are also very large. Conversely, with a small gap distance, the trajectories of the along-wind-induced responses of the two buildings have similar ranges. The along-wind-force coefficients are the same in both buildings (Figure 11), which also affects the wind-induced response.

When the gap distance is small and a link is present, the wind-induced responses of both buildings are similar, owing to the effect of the link, as shown in Figure 17. Without the link, Building 1 exhibits a positive cross-wind response, and Building 2 exhibits a negative cross-wind response. The cross-wind-force coefficients and wind-induced responses of Buildings 1 and 2 also occur in opposite directions. In this case, the two buildings can be considered to have a negative aerodynamic correlation. The results indicate that this negative correlation substantially reduces the wind-induced response due to wind loads in the LB system. The cross-wind responses occur in opposite directions in the two buildings, and a link between the buildings significantly mitigates their wind-induced responses.

Figure 18 shows that the trajectories of the wind-induced responses are relatively similar in both buildings with a large gap distance because aerodynamic characteristics (mean pressure distribution and local wind-force component coefficients) in both buildings are similar, as shown in Figure 10 and Figure 11. However, for Building 2, the wind-force coefficients are not large because Building 1 blocks it partially, and the wind-induced lateral displacements are small. Therefore, the wind-induced lateral displacements of Building 2 are smaller than those of Building 1.

When a link is present with a large gap distance, the wind-induced lateral displacements of both buildings are similar, owing to the effect of the link. Furthermore, Building 1, which exhibits a larger lateral displacement when there is no link, exhibits a larger reduction in the lateral displacement owing to the link. However, a negative correlation, as in the wind-induced response, cannot be identified with a large gap distance. Therefore, the reducing effect of the link on the lateral displacement when the gap distance is large is weaker than that on the crosswind-induced response with a small gap distance.

### 4.4. Time History of Lateral Displacement

The time history of the lateral displacement was examined to evaluate the effect of the link in detail. Figure 19 and Figure 20 show the detailed time histories of the wind-induced response during randomly selected intervals from time steps from $4\times 10^{4}$ to $4.05\times 10^{4}$.

According to the cross-wind-induced responses in Figure 19, Building 1 shows a positive response, while Building 2 shows a negative response when there is no link, i.e., a negative correlation, as mentioned in Section 4.3. Moreover, cross-wind-induced responses of the two buildings are substantially reduced when a link is installed. For the along-wind response, both buildings show positive responses, with Building 1 showing a slightly larger response. When a link is installed, the lateral displacement, according to the time history, is similar for both buildings.

When there is no link with a large gap distance, the aerodynamic characteristics (pressure distribution and wind-force coefficients) of the two buildings are similar, as shown in Figure 20. Therefore, the time histories show that both buildings have similar lateral displacements. Even when there is a link, the lateral displacements of the two tall buildings are nearly the same. However, the reduction in the lateral displacement is smaller than the reduction in the cross-wind lateral displacement with a small gap distance. These findings are considered based on the aerodynamic correlation between the two tall buildings, which again demonstrates the importance of the aerodynamic correlation based on the time history of the lateral displacement.

## 5. Conclusions

This study discussed the aerodynamic characteristics and wind-induced lateral displacements of two linked tall buildings in a staggered arrangement. Furthermore, the link effects on the lateral displacement of the two tall buildings in an LB system were also discussed. The results could facilitate the design or construction of linked tall buildings as they provide insights that can serve as reference.

With a small gap distance, a narrow jet of wind flows between the two tall buildings, and vortex shedding occurs only outside the buildings. Owing to the staggered arrangement, strong suction occurs on the inner surface of Building 1, thereby increasing the cross-wind-force coefficient and affecting the wind-induced response. Meanwhile, weaker suction acting in the opposite direction to that of Building 1 occurs on the inner surface of Building 2.

If the gap distance is large, Building 1 blocks Building 2 from the wind partially, and the shear layer generated inside Building 1 partially affects Building 2. While the pressure distribution and wind-force coefficients in the two buildings are similar, the overall magnitude is smaller in Building 2.

Comparing the lateral displacements in the two tall buildings revealed that installing a link can reduce the lateral displacement. More importantly, if the aerodynamic correlation of the wind-force components of the two buildings is negative, then because of the link, the reduction in the lateral displacement is higher than when the correlation is positive.

In the case of two buildings connected by a link, the gap distance between the two buildings has a large influence, reflecting the wind-load characteristics, and the wind load influences the wind-induced response, which is used to evaluate safety. Hence, the gap distance must be carefully considered in design and construction. As this study primarily covers wind load, further research is needed for factors such as earthquakes and soil mechanics.

## Figures and Tables

**Figure 1 sensors-21-04046-f001:**
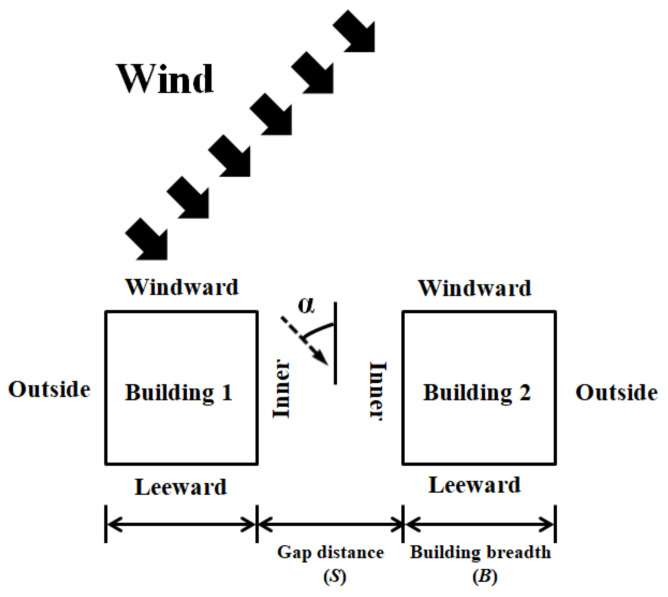
Surfaces in the building model and wind direction.

**Figure 2 sensors-21-04046-f002:**
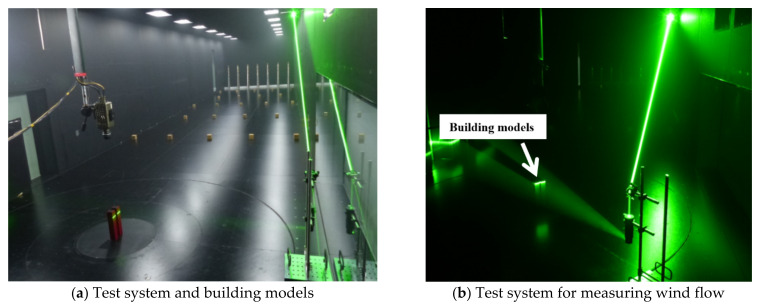
Two building models and equipment for measuring wind flow.

**Figure 3 sensors-21-04046-f003:**
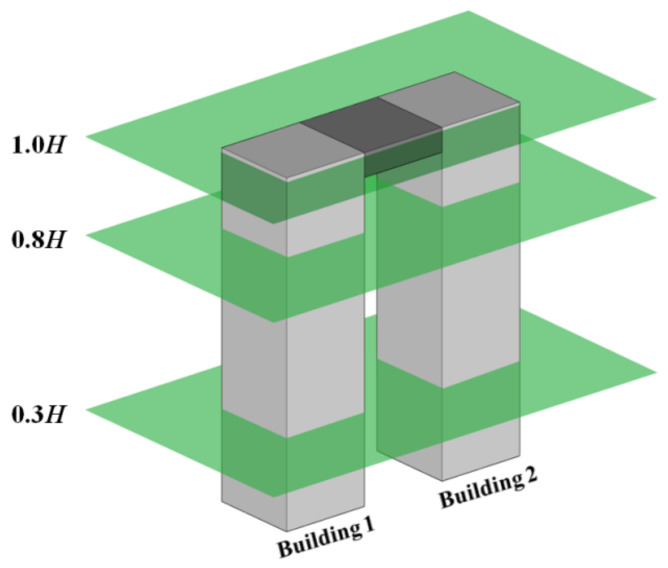
Wind-flow measurement locations.

**Figure 4 sensors-21-04046-f004:**
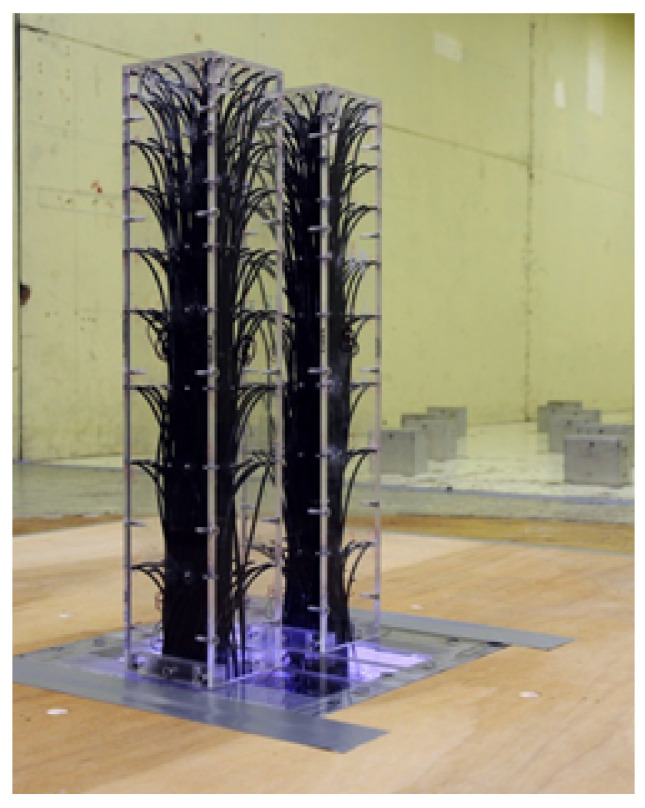
Two building models and equipment for measuring pressure.

**Figure 5 sensors-21-04046-f005:**
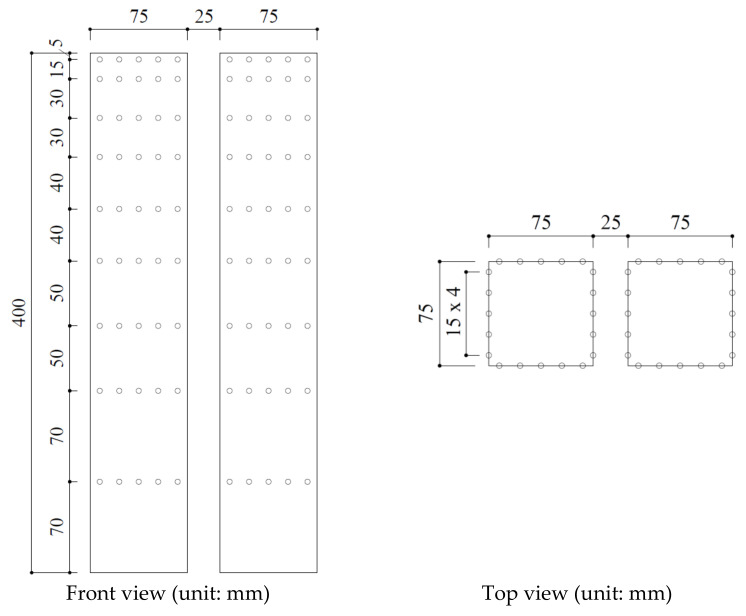
Building model size and location of pressure taps.

**Figure 6 sensors-21-04046-f006:**
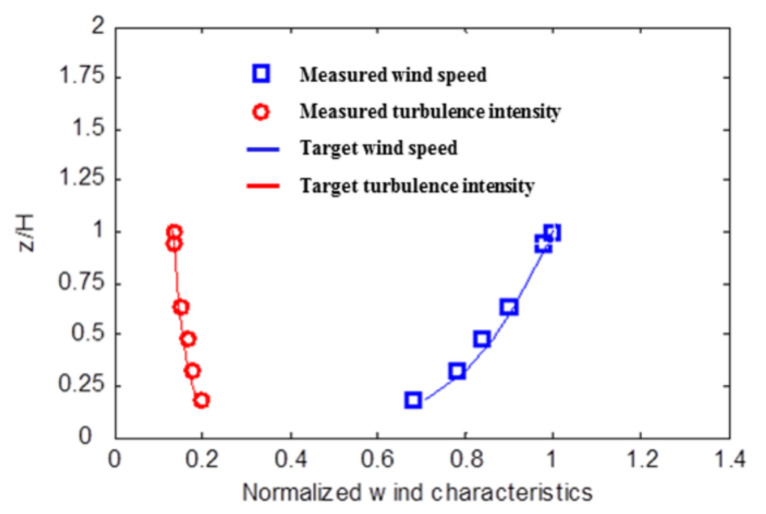
Wind profiles.

**Figure 7 sensors-21-04046-f007:**
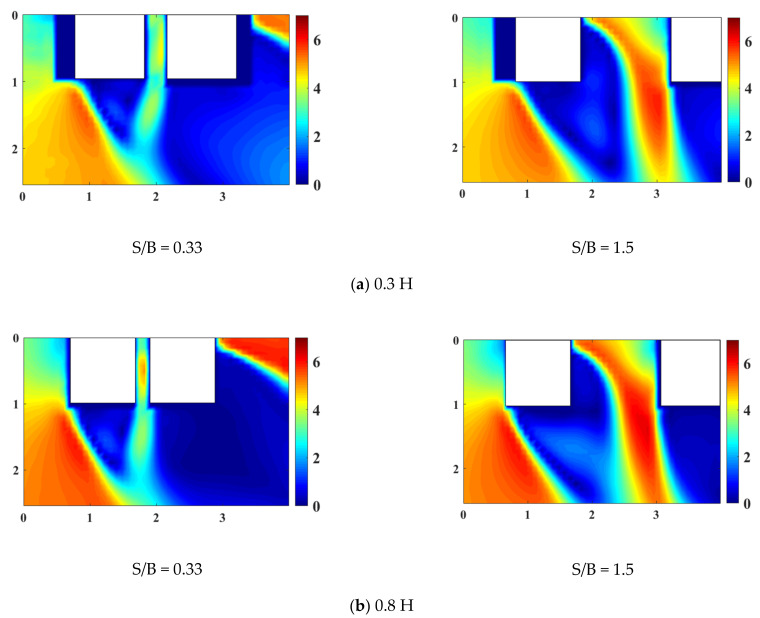
Mean velocity magnitudes around the LB model (unit: m/s).

**Figure 8 sensors-21-04046-f008:**
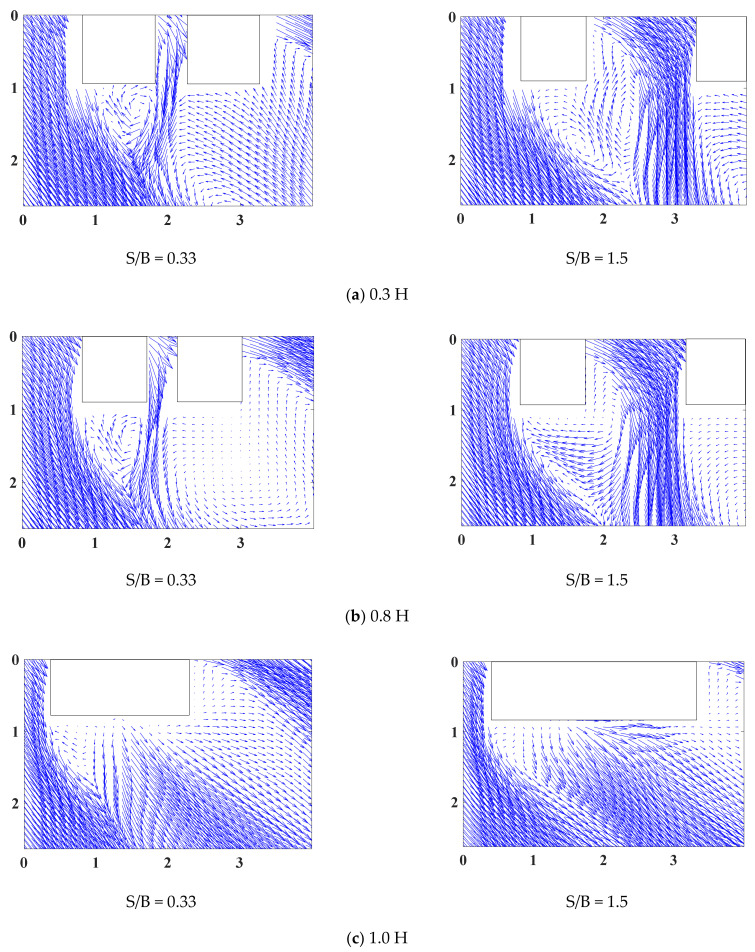
Mean velocity vectors around the LB model.

**Figure 9 sensors-21-04046-f009:**
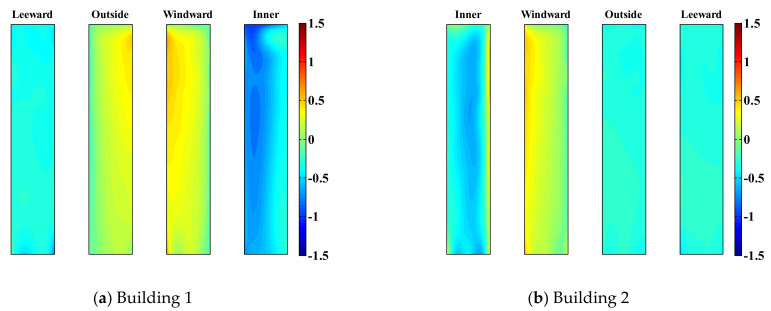
Mean pressure distributions of two buildings with a small gap distance.

**Figure 10 sensors-21-04046-f010:**
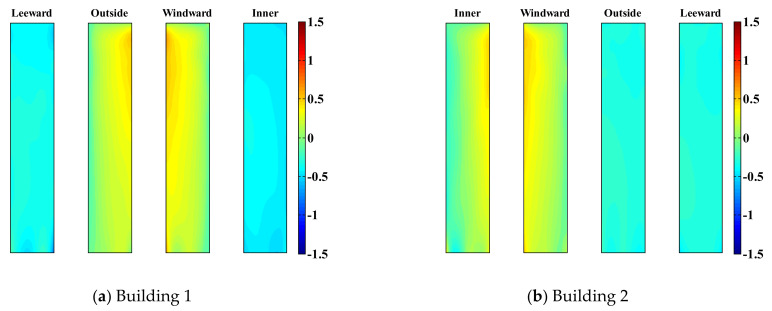
Mean pressure distributions of two buildings with a large gap distance.

**Figure 11 sensors-21-04046-f011:**
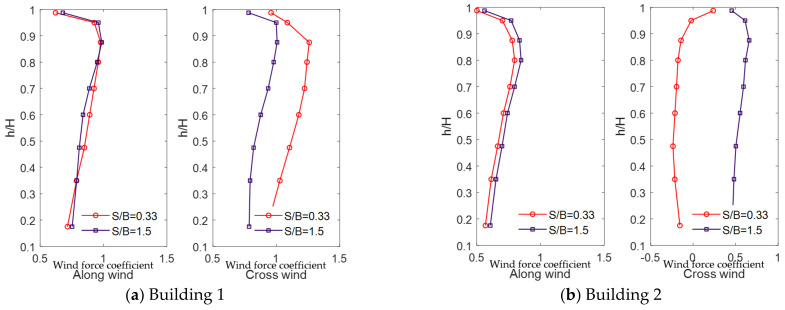
Mean values of local wind-force coefficients.

**Figure 12 sensors-21-04046-f012:**
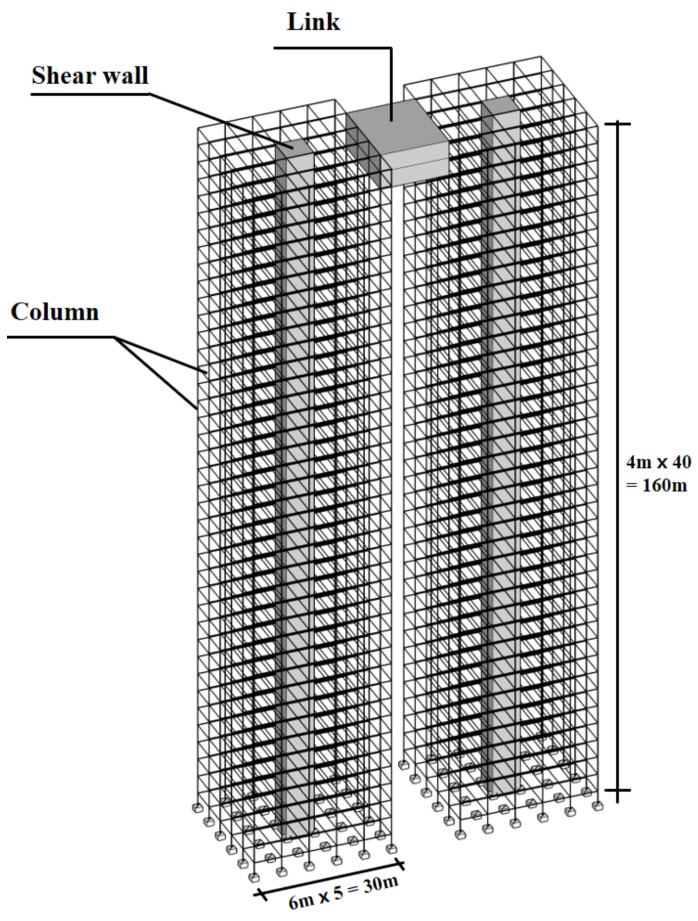
Structural system of LBs.

**Figure 13 sensors-21-04046-f013:**
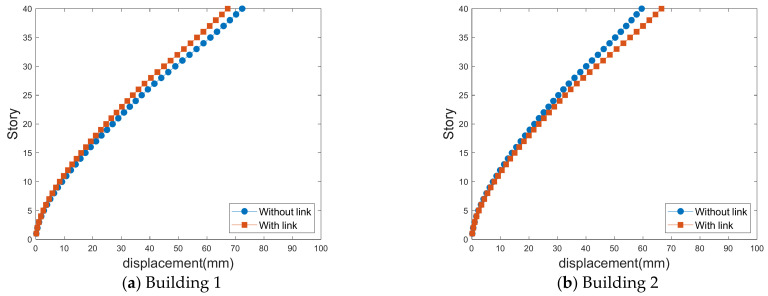
Along-wind-induced lateral displacements with a small gap distance.

**Figure 14 sensors-21-04046-f014:**
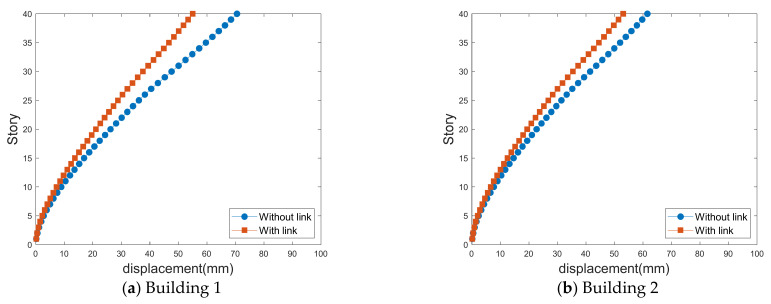
Along-wind-induced lateral displacements with a large gap distance.

**Figure 15 sensors-21-04046-f015:**
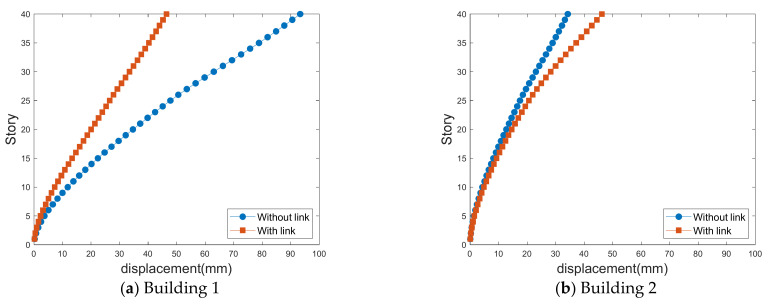
Crosswind-induced lateral displacement with a small gap distance.

**Figure 16 sensors-21-04046-f016:**
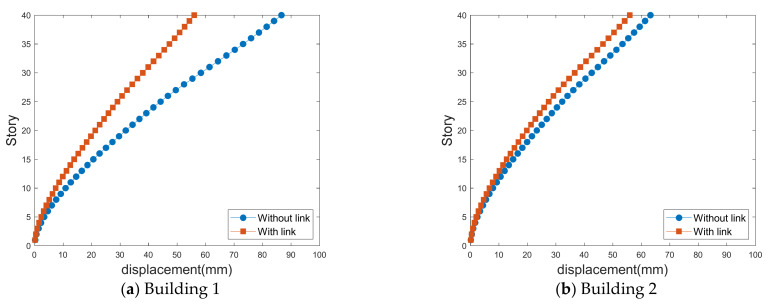
Crosswind-induced lateral displacement with a large gap distance.

**Figure 17 sensors-21-04046-f017:**
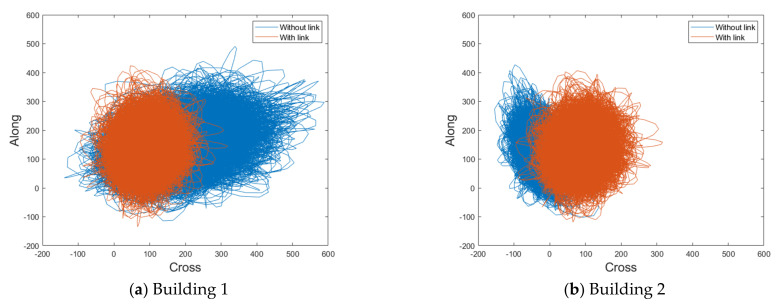
Trajectories of lateral displacement with a small gap distance (unit: mm).

**Figure 18 sensors-21-04046-f018:**
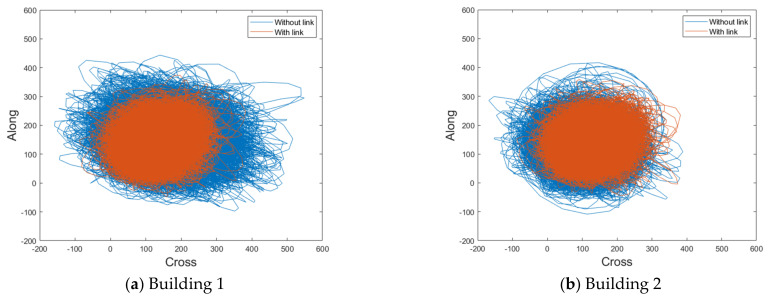
Trajectories for lateral displacement with a large gap distance (unit: mm).

**Figure 19 sensors-21-04046-f019:**
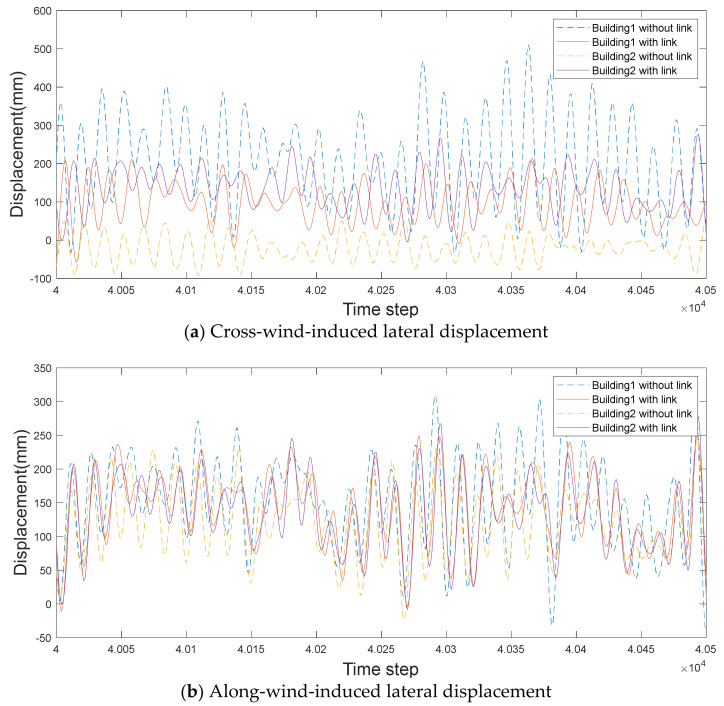
Time histories of wind-induced response with a small gap distance.

**Figure 20 sensors-21-04046-f020:**
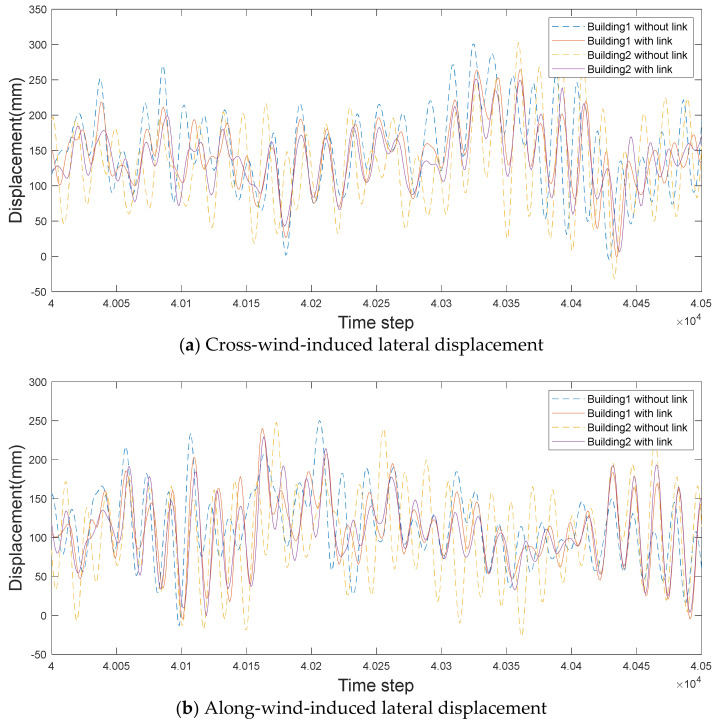
Time histories of wind-induced lateral displacements with a large gap distance.

## Data Availability

Not applicable.

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
