# Peer review of "Aerodynamic Characteristics and Lateral Displacements of a Set of Two Buildings in a Linked Tall Building System"

_sensors, 2021, doi:10.3390/s21124046_

Round 1
Reviewer 1 Report
I have read attentively this interesting manuscript. It proposes a thorough and careful study on the wind effects experienced by twin towers that can be linked in some way, to reduce the oscillations and stresses induced by wind loads.
Although my general impression is positive I would like to remark some apparent contradictions that I have detected.
In the TIle, The phrasing “two staggered buildings” seems inconclusive, perhaps it could be substituted by “a set of two buildings” or just “two parallel towers”, to avoid the repetition of the word "building".
The case study of two nearby tower buildings seems too general and as such undefined both geometrically and geographically. In a word, we do not quite know if this scheme is intended to be built, and if so, when and where. I guess that it could go in some undetermined place in Asia but if it should be for instance in Japan, seismic considerations should precede or at least be taken at the same level with this one of strong wind hindrances. In case of an earthquake over the structure the linking system could perhaps not be beneficial to avoid further damages and should be turned into a sort of damper.
If it is just a theoretical example, the dimensions should have been expressed in parametric form with some reasonable limits, of course, so that they are interchangeable to a certain extent and the results could give more universal validity.
However, if this is a theoretical exercise, we would not be able to establish the direction and intensity of the prevailing and most harmful winds and the authors should require a wider set of scenarios in order to predict damages to their structure. I do not find many diverse wind-load scenarios in this article, in fact, maybe it is my fault, but I only see the wind set at an angle of 40 degrees in the plan.
Such theoretical exercise would not be very suitable for modelling in wind tunnel facilities, because the analogy could be insufficient or inaccurate and lead to crucial errors. It is curios to outline that the wind tunnel facilities are located in Shimizu headquarters of Japan and at Hong Kong University, but anyway relatively far from the Research centres where the authors seem to develop their activities.
Another difficulty that I foresee is the need to perform several analyses by very different methods, that is: with particles, multi-pressure, flow distribution and others, because all these are sometimes not coincidental and perhaps misleading.
The wind tunnel experiments which look correct and even elegant, do not seem to correspond with any simulation software that the authors have conducted in parallel and no validation or calibration of the output of the experiment seems to be available.
In the last part of the article there comes the discussion of the LB and this is preceded by more theoretical formulation of the matrix theory for structures. It remains unclear how these matrixes and stiffness and displacements connect with the former analyses of wind pressures.
Based on all the former, the conclusions could be slightly more specific and focus in more detail on the findings and future design recommendations. Other considerations like earthquakes and soil mechanics and resistance should be mentioned at least.
Reviewer 2 Report
In this paper, the aerodynamic characteristics and lateral displacements of a linked tall building system (LB system) in staggered arrangement are studied. And two wind tunnel tests that using particle image velocimetry and multi-pressure measurement system are conducted to investigate the wind flow and wind loads. Furthermore, based on the experimental results, a 3-dimensional analysis is also applied to evaluate the influence of the link in this LB system. It is no wonder that the paper has some innovation and engineering value. And the work of this paper is very comprehensive. However, the paper still needs a deep and accurate revision before it could be considered for publication.
Detailed comments:
- The introduction section of the current paper should be improved for the following reasons:
(1) The references in the introduction are too old to reflect the research status. More up to date references should be added to the literature review.
(2) In Line 68, the author said that previous studies mainly focused on investigating the aerodynamic characteristics of cylinders in the case of staggered arrangements rather than typical buildings. However, only the references from 15 years ago are used to support this statement. In fact, there are also many studies about typical buildings in staggered arrangements in recent research. So the author needs to clarify this point further.
(3) The fourth paragraph of the introduction presents the studies conducted on the lateral displacement of the LB system. Why the reference [24] about structures in the absence of a link was applied in line 73 and 74?
- The main purpose of the paper is to reveal the aerodynamic characteristics and lateral displacements of LB systems in staggered arrangement. However, whether in the test or simulation, the selected buildings are still arranged side by side, while the input direction of the wind excitation is rotated by an angle. More explanation should be provided to demonstrate whether this approach is reasonable.
- The models of the experimental buildings cannot be observed in Figure 2.
- In section 2.2, the MPMS, the corresponding structural model and the test method in the wind tunnel experiment are same as those in reference [25]. And in section 4, the 3D analysis also uses the same structural model in reference [25]. Therefore, more explanation should be provided to clarify the main novelty of the manuscript.
- Figure 6 in this manuscript is consistent with the figure 9 in the reference [25], but why the ordinate (z/H) in figure 6 can greater than 1?
- The specific name of the abscissa in figure 11 is absent.
- More details of the calculated LB system should be implemented in the 3D analysis section, such as the dynamic characteristics of the system.
- The 3D analysis of prototype LB system is conducted based on the wind tunnel experiments of the scaled structures in section 2.2. However, when introducing the experiments, only the similarity coefficient in dimension have been presented, which is insufficient to complete the mapping relationship between the test and the prototype structures. Hence, more details of the experiments should be presented.
- It is better to use different name for I’ in line 387 and I in line 388.
- The symbol of the shear deformation constant in line 387 is incorrect.
- How to obtain the comment in line 436 according to figure 15 and 16?
In sum, this paper has some innovation, while a deep and accurate revision still required. Therefore, the suggestion is major revision and re-review.
Round 2
Reviewer 2 Report
In this paper, the aerodynamic characteristics and lateral displacements of a linked tall building system (LB system) in staggered arrangement are studied. And two wind tunnel tests that using particle image velocimetry and multi-pressure measurement system are conducted to investigate the wind flow and wind loads. Furthermore, based on the experimental results, a 3-dimensional analysis is also applied to evaluate the influence of the link in this LB system. This paper has some innovations and engineering values. In the revised paper, the author answers some comments of reviewers and modify the paper by the suggestion. However, there are some unclear points need better explanation.
Detailed comments:
- It can be seen from figure 1, the test buildings are still arranged side-by-side instead of staggered, where the inner wall of the two buildings are facing directly and the link is perpendicular to them. This arrangement may not correspond with the actual project. However, in the revision, there is no proper explanation to this.
- It better to supplement one more photo about PIV test in figure 2, which is clearer and brighter and the whole test system including the buildings, the laser is visible.
3. In figure 15, the displacements of building 2 is obviously smaller than building 1 when the link is absent, However, the displacements of the two buildings are similar when the link is added. Thus, why the author said in page 18 and 19 (revised paper) that “the cross-wind response of Building 2 was reduced to a greater extent than that of Building 1, when a link was present”
